# Morphological and phylogenetic appraisal of *Ophioceras* (Ophioceraceae, Magnaporthales)

**Hong-Bo Jiang**[1,2,3,4,5], **Kevin D. Hyde**[1,2,3,6,7], **Er-Fu Yang**[2,5,8,9], **Pattana Kakumyan**[4], **Ali H. Bahkali**[6], **Abdallah M. Elgorban**[6], **Samantha C. Karunarathna**[2,5,10], **Rungtiwa Phookamsak**[1,2,5,10]*, **Saisamorn Lumyong**[1,8,11]*

**1** Faculty of Sciences, Research Center of Microbial Diversity and Sustainable Utilization, Chiang Mai University, Chiang Mai, Thailand, **2** CAS Key Laboratory for Plant Diversity and Biogeography of East Asia, Kunming Institute of Botany, Chinese Academy of Sciences, Kunming, Yunnan, P.R. China, **3** Center of Excellence in Fungal Research, Mae Fah Luang University, Chiang Rai, Thailand, **4** School of Science, Mae Fah Luang University, Chiang Rai, Thailand, **5** Honghe Center for Mountain Futures, Kunming Institute of Botany, Chinese Academy of Sciences, Honghe County, Yunnan, P.R. China, **6** Department of Botany and Microbiology, College of Science, King Saud University, Riyadh, Saudi Arabia, **7** Innovative Institute for Plant Health, Zhongkai University of Agriculture and Engineering, Haizhu District, Guangzhou, Guangdong, P.R. China, **8** Faculty of Science, Department of Biology, Chiang Mai University, Chiang Mai, Thailand, **9** Faculty of Science, Master of Science Program in Applied Microbiology (International Program), Chiang Mai University, Chiang Mai, Thailand, **10** CIFOR-ICRAF China Program, World Agroforestry (ICRAF), Kunming, Yunnan, China, **11** Academy of Science, The Royal Society of Thailand, Bangkok, Thailand

* jomjam.rp2@gmail.com (RP); scboi009@gmail.com (SL)

**Data Availability Statement:** All relevant data are within the paper and its Supporting information files.

## Abstract

*Ophioceras* is accommodated in the monotypic family Ophioceraceae (Magnaporthales, Sordariomycetes), and the genus is delimited based on molecular data. During an ongoing survey of bambusicolous fungi in southwest China, we collected a submerged decaying branch of bamboo from Sichuan Province, China and an *Ophioceras* species occurring on this substrate was observed and isolated. An *Ophioceras* taxon was delimited based on morphological characteristics and combined LSU, RPB1 and ITS sequence analyses and is described as *Ophioceras sichuanense* sp. nov. The species formed a well-supported clade basal to *Ophioceras* (100% ML, 1.00 PP). Based on the updated phylogenetic tree of Magnaporthales, *Ceratosphaerella castillensis* (generic type) and *C. rhizomorpha* formed a clade within *Ophioceras* and morphologically resemble *Ophioceras*. Therefore, *Ceratosphaerella* is synonymized under *Ophioceras*. The phylogenetic relationships of *Ophioceras* are discussed in relation to morphological similarities of genera in Magnaporthales. The generic circumscription of *Ophioceras* is emended.

## Introduction

Klaubauf et al. [1] introduced the family Ophioceraceae to accommodate *Ophioceras* Sacc. which is typified by *O. dolichostomum* (Berk. & M.A. Curtis) Sacc. The family is recognized as black, immersed to superficial, globose to subglobose, perithecial ascomata with long, periphysate necks, 8-spored, unitunicate, subcylindrical to narrowly fusoid asci with a J-, apical ring,

**Funding:** The authors are grateful for the support of the Mushroom Research Foundation, Chiang Rai, Thailand (to H-BJ) and Yunnan Provincial Science and Technology Department grant no. 202003AD150004 (to RP and H-BJ under Jianchu Xu). KDH thanks the Foreign Experts Bureau of Yunnan Province, Foreign Talents Program (2018; grant no. YNZ2018002), Impact of Climate Change on Fungal Diversity and Biogeography in the Greater Mekong Subregion (grant no: RDG6130001). RP sincerely acknowledges the CAS President's International Fellowship Initiative (PIFI) for young staff (grant no. Y9215811Q1), the "High-level Talent Support Plan" Young Top Talent Special Project of Yunnan Province and Chiang Mai University for financial support. SCK thanks the CAS President's International Fellowship Initiative (PIFI) young staff under the grant number: 2020FYC0002 and the National Science Foundation of China (NSFC) under the project code 31851110759 for funding. AHB and AME thank the International Scientific Partnership Program ISPP at King Saud University through ISPP#0089. The funders had no role in study design, data collection and analysis, decision to publish, or preparation of the manuscript.

**Competing interests:** The authors have declared that no competing interests exist.

and filiform, hyaline to olivaceous, septate ascospores without sheaths [1–3]. Species of Ophioceraceae are saprobes on wood and herbaceous plants in aquatic or terrestrial habitats [1–6].

Saccardo [7] introduced *Ophioceras* to accommodate taxa with immersed, sub carbonaceous, globose perithecia, with conical-cylindrical, filiform ostioles, elongate asci and filiform, septate ascospores. The genus initially accommodated *O. bacillatum* (Cooke) Sacc., *O. dolichostomum*, *O. friesii* (Mont.) Sacc., *O. hystrix* (Ces.) Sacc., *O. macrocarpum* (Sacc.) Sacc., *O. longisporum* Sacc. and *O. therryanum* (Sacc. & Roum.) Sacc [7]. *Ophioceras* was accommodated in Magnaporthaceae based on limited molecular data [8–10]. However, the genus was excluded from Magnaporthaceae and accommodated in the separate family Ophioceraceae in Magnaporthales based on combined LSU and RPB1 phylogenetic analyses [1]. To date, the asexual morph of *Ophioceras* has not yet been reported [1,2,6], and only 40 epithets are listed under this genus in Index Fungorum [11]. However, only eleven *Ophioceras* species have molecular data in GenBank, and only LSU and ITS sequence data are most available for these species [12].

*Ophioceras* species are commonly discovered from wood in freshwater and are generally clumped under the name *Ophioceras* sp. [13,14], *O. commune* [15–17] or *O. dolichostomum* [18–21]. Twenty-three *Ophioceras* species are listed in Species Fungorum [22] and accepted in Hyde et al. [3]. *Ophioceras* species occur in disparate streams in different islands and continents and are therefore likely to have separated for millions of years ago, potentially explaining their evolution into distinct species. The number of existing *Ophioceras* species is therefore likely to exceed more than presently known. Hyde et al. [23] has shown that numerous new taxa await description across most under-collected and -studied.

*Ceratosphaerella* comprises *C. castillensis* (C.L. Sm.) Huhndorf, Greif, Mugambi & A.N. Mill. and *C. rhizomorpha* Huhndorf & Mugambi, introduced by Huhndorf et al. [8]. Phylogenetic analyses of the LSU sequence dataset showed that *Ceratosphaerella* grouped with *Ophioceras* in Magnaporthaceae [8]. Klaubauf et al. [1] only accommodated *Ophioceras* into Ophioceraceae, but they did not incorporate *Ceratosphaerella* in their analyses. Thus, *Ceratosphaerella* remains in Magnaporthaceae. To date, there are only two species included in *Ceratosphaerella* [11], and these taxa only have LSU sequence data in GenBank [12]. *Ceratosphaerella castillensis* has been reported only as a sexual morph, while *C. rhizomorpha* is holomorphic and has a Didymobotryum-like asexual morph [8]. In Luo et al. [2], *Ceratosphaerella* did not group within Magnaporthaceae but clustered with *Ophioceras* in Ophioceraceae. However, Luo et al. [2] did not verify the phylogenetic status of *Ceratosphaerella* in Ophioceraceae, leading to the uncertain placement of the genus.

This study aims to introduce the novel *Ophioceras* taxon on a submerged bamboo branch in Sichuan Province, China and resolve the congeneric status of *Ceratosphaerella* and *Ophioceras* in Ophioceraceae based on a morpho-molecular approach.

## Materials and methods

### Sample collection, observation and isolation

Decaying branches of bamboo submerged in freshwater were collected in the stream in the Shunan Artificial Bamboo Forest, Sichuan Province, P.R. China in July 2019. Samples were kept in a paper bag for further morphological examination in the laboratory. Pseudostromata visualized on decaying branches of bamboo were observed and examined under a stereo microscope (Motic series SMZ 140) and captured via digital phone camera (iPhone 7, Apple Inc., USA). Microscopic features (e.g., asci, ascospores and paraphyses) were prepared using the squashing mount technique in sterilized distilled water on clean slides for morphological study. Sections of pseudostromatic ascomata, ostiolar necks and peridial structures were by

free-hand sectioning using Gillette razor blades. Melzer's reagent and Indian ink were used to detect the J-/J+ apical ring of the asci and mucilaginous sheath surrounding the ascospores, respectively. Morphological features visualized under a Nikon ECLIPSE Ni compound microscope were photographed using a Canon EOS 600D digital camera. Measurements (n = 10–20) of pseudostromata, locules, peridium, paraphyses, asci and ascospores were carried out using Tarosoft (R) Image Frame Work version 0.9.7. Photographic plate and line drawings of fungal morphologies were edited and combined using Adobe Photoshop CS6 (Adobe Systems Inc., USA).

Single spore isolation based on the spore suspension technique [24] was carried out to obtain a pure fungal culture. Germinated ascospores were transferred to the new potato dextrose agar plates (PDA; Qingdao Daily Water Biotechnology co. LTD. Shandong, P.R. China) under aseptic conditions and grown under normal day/nightlight at room temperature. Culture characteristics (e.g., growth, shape, colour, margin, elevation, consistency) were checked and recorded after one week and four weeks.

The holotypic specimen is conserved in the herbarium of Cryptogams Kunming Institute of Botany Academia Sinica (KUN-HKAS), Yunnan, P.R. China. The isotype is stored in the herbarium of Mae Fah Luang University, Chiang Rai, Thailand (MFLU). Ex-type living cultures are preserved in the Kunming Institute of Botany Culture Collection (KUMCC) and Mae Fah Luang University Culture Collection (MFLUCC). Facesoffungi and Index Fungorum numbers were registered for the new taxon [11,25].

## DNA extraction, amplification and sequencing

Fungal genomic DNA was extracted from fresh mycelia using the Biospin Fungus Genomic DNA Extraction Kit (BioFlux®, P.R. China) following manufacturer's instructions (Hangzhou, P.R. China) and also extracted from fruiting bodies (= pseudostromata) directly using the Forensic DNA Kit (Omega®, USA) for a duplicated strain. DNA amplification was performed by polymerase chain reaction (PCR). Two gene regions including the internal transcribed spacer (ITS) and 28S large subunit rDNA (LSU), were used to amplify PCR fragments using forward and reward primer pairs: ITS5/ITS4 [26] and LR0R/LR5 [27], respectively. PCR reactions were conducted in a 25 μL total volume, consisted of 2 μl of DNA template, 1 μl of each forward and reverse primer, 12.5 μl of 2× Power Taq PCR Master Mix (Beijing BioTeke Corporation, P.R. China) and 8.5 μl double-distilled water (ddH$_2$O). The PCR thermal cycle program for ITS and LSU was set up following Jiang et al. [28]. PCR fragments were purified and sequenced at TsingKe Biological Technology (Beijing) Co., Ltd, P.R. China.

## Molecular phylogeny

The newly generated sequences (ITS and LSU) of fungal strains were initially subjected to the basic local alignment search tool (BLASTn) via the National Center for Biotechnology Information web portal (NCBI; https://blast.ncbi.nlm.nih.gov) for discovering closely related fungal taxa. In order to clarify the phylogenetic placement of the new isolate, the representative taxa in Magnaporthales were incorporated with the new taxon to generate the sequence data matrix for further analysis. These representative taxa of Magnaporthales were downloaded from the GenBank database (Table 1) based on recent publications [2,29].

Preliminary single-gene data matrixes were aligned via MAFFT v. 7.452 [30] and improved manually in BioEdit v. 5.0.6 [31]. The single-gene alignments of LSU and ITS data matrixes were prior analyzed by maximum-likelihood (ML) criterion using RAxML v. 7.0.3 [32,33] for checking if there are any conflicts between the tree topologies. The concatenated LSU-ITS and LSU-RPB1-ITS sequence datasets were further analyzed based on maximum-likelihood (ML)

**Table 1. Detailed information of fungal taxa used in the phylogenetic analyses.** The newly generated sequences are indicated in **blue**, and the ex-type strains are in **bold**.

| Species name | Culture collection/ Voucher no. | GenBank accession numbers | | |
|---|---|---|---|---|
| | | LSU | RPB1 | ITS |
| *Aquafiliformis lignicola* | **MFLUCC 16–1341** | MK835815 | / | MK828615 |
| *Aquafiliformis lignicola* | MFLUCC 18–1338 | MK835814 | / | MK828614 |
| *Bambusicularia brunnea* | **INA-B-92-45** | NG_058671 | KM485043 | NR_145387 |
| *Barretomyces calatheae* | CBS 129274 | MH876639 | KM485045 | MH865202 |
| *Bifusisporella sorghi* | **URM 7442** | NG_067852 | MK060159 | NR_164042 |
| *Budhanggurabania cynodonticola* | **BRIP 59305** | NG_058678 | KP162143 | NR_137952 |
| *Buergenerula spartinae* | ATCC 22848 | DQ341492 | JX134720 | JX134666 |
| *Bussabanomyces longisporus* | **CBS 125232** | NG_058668 | KM485046 | NR_145385 |
| *Ceratosphaeria aquatica* | **MFLU 18–2323** | MK835812 | / | MK828612 |
| *Ceratosphaeria lampadophora* | SMH 4822 | AY346270 | / | / |
| *Ceratosphaeria lignicola* | **MFLU 18–1457** | MK835813 | / | MK828613 |
| *Deightoniella roumeguerei* | CBS 128780 | MH876533 | KM485047 | MH865092 |
| *Falciphora oryzae* | **CBS 125863** | NG_064356 | KJ026706 | NR_153972 |
| *Falciphoriella solaniterrestris* | **CBS 117.83** | NG_058108 | KM485058 | NR_153995 |
| *Gaeumannomycella caricis* | **CBS 388.81** | NG_058109 | KM485059 | NR_146245 |
| *Gaeumannomyces amomi* | CMUZE002 | DQ341493 | / | AY265318 |
| *Gaeumannomyces radicicola* | **CBS 296.53** | NG_058089 | KM009194 | NR_146246 |
| *Gaeumannomyces tritici* | CBS 541.86 | DQ341497 | / | / |
| *Kohlmeyeriopsis medullaris* | **CBS 117849** | NG_058110 | KM485068 | NR_154068 |
| *Macgarvieomyces borealis* | **CBS 461.65** | NG_058088 | KM485070 | NR_145384 |
| *Macgarvieomyces juncicola* | CBS 610.82 | KM484970 | KM485071 | KM484855 |
| *Magnaporthiopsis incrustans* | M35 | JF414892 | JF710437 | JF414843 |
| *Magnaporthiopsis maydis* | CBS 133165 | KX306614 | / | KX306544 |
| *Magnaporthiopsis maydis* | **CBS 662.82A** | NG_058111 | KM485072 | NR_154175 |
| *Magnaporthiopsis poae* | **M47** | JF414885 | JF710433 | JF414836 |
| *Muraeriata africana* | **GKM 1084** | EU527995 | / | / |
| *Muraeriata collapsa* | **SMH 4553** | EU527996 | / | / |
| *Myrmecridium schulzeri* | CBS 100.54 | EU041826 | / | EU041769 |
| *Myrmecridium sorbicola* | **CBS 143433** | NG_063957 | / | NR_158871 |
| *Nakataea oryzae* | CBS 252.34 | MH867001 | KM485078 | KM484862 |
| *Nakataea oryzae* | CBS 288.52 | MH868571 | KM485080 | MH857040 |
| *Neocordana malayensis* | **CBS 144604** | NG_066327 | / | NR_163364 |
| *Neocordana musae* | **CPC 18127** | LN713290 | / | NR_154265 |
| *Neogaeumannomyces bambusicola* | **MFLUCC 11–0390** | NG_059556 | / | NR_146247 |
| *Neopyricularia commelinicola* | **CBS 128308** | NG_058112 | KM485087 | NR_154226 |
| *Omnidemptus affinis* | **ATCC 200212** | NG_059478 | JX134728 | NR_154292 |
| *Omnidemptus graminis* | **CBS 138107** | MK487734 | / | NR_164058 |
| *Ophioceras aquaticus* | **IFRDCC 3091** | JQ797433 | / | JQ797440 |
| *Ophioceras aquaticus* | MFLUCC 16–0906 | MK835810 | / | MK828611 |
| *Ophioceras castillensis* (as *Ceratosphaeria castillensis*) | SMH 1865 | EU527997 | / | / |
| *Ophioceras chiangdaoense* | **CMU 26633** | NG_066356 | / | / |
| *Ophioceras commune* | KUN-HKAS 92569 | MH795820 | / | MH795815 |
| *Ophioceras commune* | KUN-HKAS 92587 | MH795819 | / | MH795814 |
| *Ophioceras commune* | KUN-HKAS 92590 | MK835809 | / | MK828610 |
| *Ophioceras commune* | KUN-HKAS 92640 | MH795818 | / | MH795813 |

(*Continued*)

**Table 1.** (*Continued*)

| Species name | Culture collection/ Voucher no. | GenBank accession numbers | | |
|---|---|---|---|---|
| | | LSU | RPB1 | ITS |
| *Ophioceras dolichostomum* | CMURp50 | DQ341504 | / | / |
| *Ophioceras hongkongense* | HKUCC 3624 | DQ341509 | / | / |
| **Ophioceras leptosporum** | **CBS 894.70** | NG_057959 | JX134732 | NR_111768 |
| **Ophioceras rhizomorpha** (as *Ceratosphaerella rhizomorpha*) | **GKM 1262** | EU527998 | / | / |
| *Ophioceras sichuanense* | KUN-HKAS 107677 | MW057782 | / | MW057779 |
| **Ophioceras sichuanense** | **KUMCC 20–0213** | MT995046 | / | MT995045 |
| **Ophioceras submersum** | **MFLUCC 18–0211** | MK835811 | / | / |
| **Proxipyricularia zingiberis** | **CBS 133594** | NG_063934 | KM485091 | AB274434 |
| **Pseudohalonectria fagicola** | **MFLUCC 15–1117** | KX426219 | / | / |
| **Pseudohalonectria hampshirensis** | **MFLUCC 15–0774** | KX426218 | / | / |
| *Pseudohalonectria lignicola* | SMH 2440 | AY346299 | / | / |
| *Pseudohalonectria lutea* | CBS 126574 | MH875622 | / | MH864160 |
| **Pseudophialophora eragrostis** | **CM12m9** | KF689638 | KF689618 | NR_146240 |
| **Pseudopyricularia cyperi** | **CBS 133595** | NG_058113 | / | NR_137920 |
| **Pseudopyricularia kyllingae** | **CBS 133597** | NG_058114 | KM485096 | NR_155645 |
| *Pyricularia ctenantheicola* | GR0001 | KM484994 | KM485098 | KM484878 |
| *Pyricularia grisea* | BR0029 | KM484995 | KM485100 | KM484880 |
| *Pyriculariopsis parasitica* | HKUCC 5562 | DQ341514 | / | / |
| **Slopeiomyces cylindrosporus** | **CBS 609.75** | KM485040 | KM485158 | KM484944 |
| *Slopeiomyces cylindrosporus* | CBS 610.75 | NG_057751 | JX134721 | NR_120170 |
| *Xenopyricularia zizaniicola* | CBS 132356 | KM485042 | KM485160 | KM484946 |

**Abbreviations**: **ATCC**: American Type Culture Collection, Virginia, USA; **CBS**: Westerdijk Fungal Biodiversity Institute, Utrecht, Netherlands; **CMU**: Chiang Mai University, Chiang Mai, Thailand; **CPC**: Culture Collection of Pedro Crous, Netherlands; **KUN-HKAS**: the Herbarium of Cryptogams Kunming Institute of Botany Academia Sinica, Yunnan, P.R. China; **HKUCC**: Hong Kong University Culture Collection, Hong Kong, P.R. China; **IFRDCC**: Fungal Research & Development Centre Culture Collection, P.R. China; **KUMCC**: Kunming Culture Collection, Yunnan, P.R. China; **MFLU**: the Herbarium of Mae Fah Luang University, Chiang Rai, Thailand; **MFLUCC**: Mae Fah Luang University Culture Collection, Chiang Rai, Thailand.

and Bayesian inference (BI) criteria and the tree topologies of these combined gene datasets were compared for checking the congruence of the tree topologies. The concatenated LSU-ITS and LSU-RPB1-ITS sequence datasets comprise 64 strains of ingroup taxa in Magnaporthales. *Myrmecridium schulzeri* (CBS 100.54) and *M. sorbicola* (CBS 143433) were selected as the out-group taxa.

Maximum-likelihood (ML) criterion was analyzed by the online tool RAxML-HPC v.8 on XSEDE (8.2.12) via CIPRES Science Gateway v. 3.3 web portal [34]. The ML + thorough boot-strap parameters were set at default values but modified as 1000 replications of bootstraps (-N 1000) and using the GTRGAMMAI model.

The best-fit evolutionary models of nucleotide substitution for LSU, RPB1 and ITS loci were evaluated by MrModeltest 2.3 [35], and the GTR+I+G substitution model under the Akaike Information Criterion (AIC) was the best-fit evolutionary model for each locus. Bayes-ian inference (BI) analysis was performed by MrBayes v. 3.1.2 [36]. The Markov Chain Monte Carlo sampling (MCMC) sampling method was used to determine posterior probabilities (PP) [37,38]. One million generations of six simultaneous Markov chains were run and sampled every 100th generation. MCMC heated chain was set up with a "temperature" value at 0.15. The burn-in was set to 20% of all sampled trees, meaning that sampled trees beneath the

asymptote (20%) were discarded. Posterior probabilities values were then calculated from the remaining 8000 trees in the majority rule consensus tree.

The final phylogram presented in this study was visualized in FigTree v. 1.4.0 (http://tree. bio.ed.ac.uk/software/figtree/). The phylogenetic tree was edited in Microsoft Office Power-Point 2016 (Microsoft Inc., USA) and converted to jpeg file using Adobe Photoshop CS6 (Adobe Systems Inc., USA). New sequences generated from the present study were registered for GenBank accession numbers (Table 1). The final alignment and phylogram are submitted in TreeBASE submission ID: 28293 (http://purl.org/phylo/treebase/phylows/study/TB2: S28293?x-access-code=66338d666c9ae6b7c0a0aa779b50078d&format=html).

## Nomenclature

The electronic version of this article in Portable Document Format (PDF) in a work with an ISSN or ISBN will represent a published work according to the International Code of Nomenclature for algae, fungi, and plants, and hence the new names contained in the electronic publication of a PLOS ONE article are effectively published under that Code from the electronic edition alone, so there is no longer any need to provide printed copies.

In addition, new names contained in this work have been submitted to Index Fungorum from where they will be made available to the Global Names Index. The unique Index Fungorum number can be resolved, and the associated information viewed through any standard web browser by appending the Index Fungorum number contained in this publication to the prefix www.indexfungorum.org/. The online version of this work is archived and available from the following digital repositories: PubMed Central and LOCKSS.

## Compliance with ethical standards

There is no conflict of interest (financial or non-financial) and all authors have agreed to submission of paper. The authors also declare that they have no conflict of interest and confirm that the field studies did not involve endangered or protected species.

## Results

### Molecular phylogeny

Based on the results from the nucleotide BLAST search tool of LSU sequence, our new strains (KUMCC 20–0213 and KUN-HKAS 107677) are closely related to species of *Ophioceras*, whereas ITS sequence revealed that our new strains are similar to the unidentified fungal endophyte isolate 4583 (86.79% similarity) and other taxa in Magnaporthales. The concatenated LSU-RPB1-ITS dataset included 2594 total characters with gaps (LSU: 1–905 bp, RPB1: 906–1877 bp, ITS: 1878–2594 bp). The best scoring ML tree with the final ML optimization likelihood value of -23570.931644 (ln) was selected to represent the phylogenetic relationships of taxa in Magnaporthales (Fig 1). All free model parameters were estimated using the GTRGAMMAI model, with 1354 distinct alignment patterns and 40.89% of undetermined characters or gaps. Estimated base frequencies were as follows: A = 0.247576, C = 0.255381, G = 0.292293, T = 0.204750, with substitution rates AC = 1.508478, AG = 2.781197, AT = 1.835341, CG = 0.983246, CT = 6.427316, GT = 1.000000. The Tree-Length = 7.381586 and the gamma distribution shape parameter $\alpha$ = 0.605644. The evaluation of Bayesian posterior probabilities (BYPP) from MCMC was carried out with the final average standard deviation of split frequencies reached 0.009301.

The tree topology from ML analysis showed similar results with the BI analysis and comparing LSU-ITS, and LSU-RPB1-ITS phylograms also revealed similarities in overall

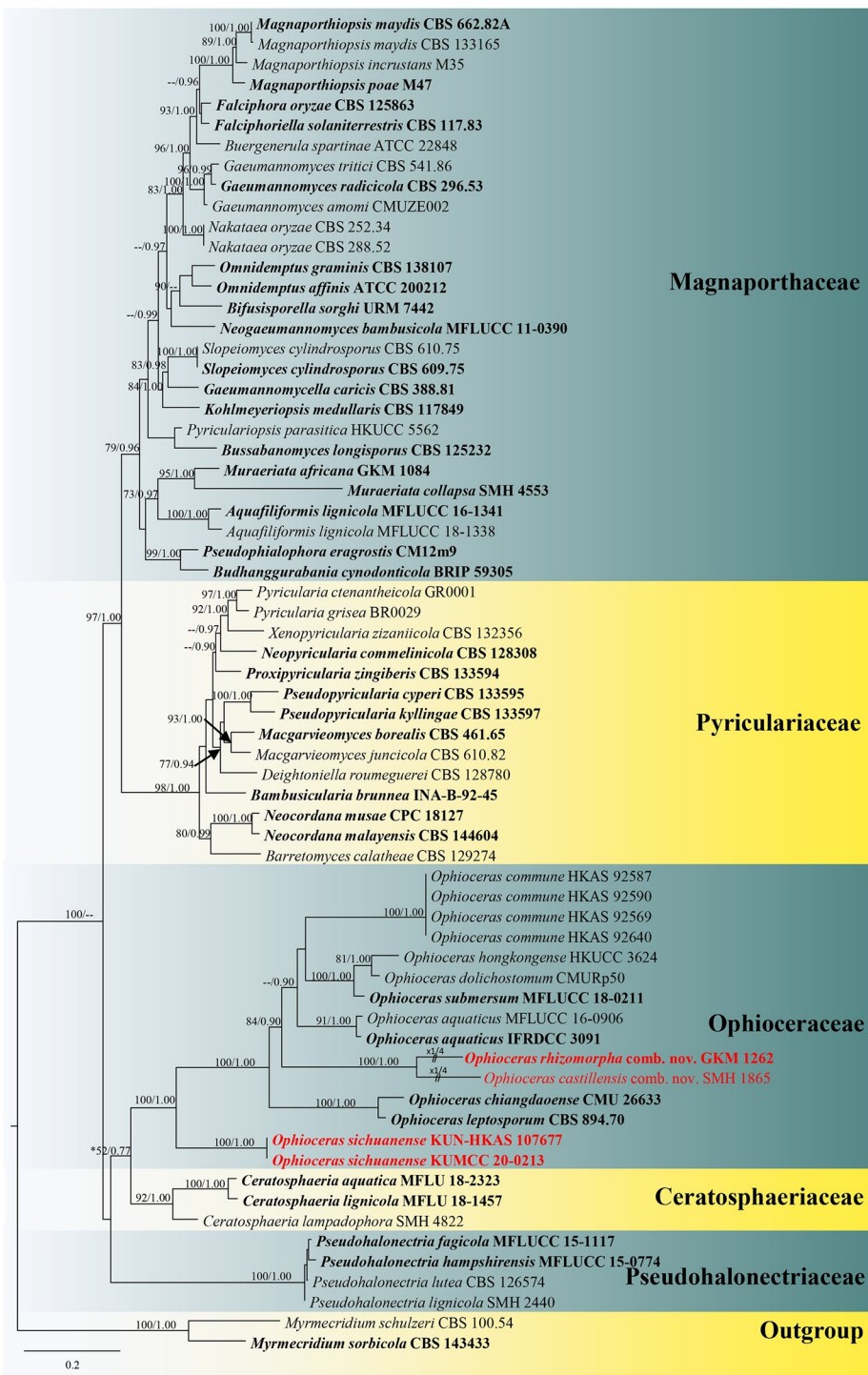

**Fig 1. Maximum likelihood tree based on a combined LSU, RPB1 and ITS sequence matrix for taxa in Magnaporthales.** Bootstrap support values for ML equal to or higher than 70% and the Bayesian posterior probabilities equal to or higher than 0.90 PP are defined above the nodes as ML/PP. Ex-type strains are in black bold, and new species and new combinations are indicated in red bold.

topologies (Fig 1 and S2 Fig). Thus, we will use the LSU-RPB1-ITS topology for further discussion. Five families of Magnaporthales were included in the presented phylogenetic analyses viz. Ceratosphaeriaceae, Magnaporthaceae, Ophioceraceae, Pseudohalonectriaceae and Pyriculariaceae. These five families formed well-resolved monophyletic clades within Magnaporthales with significant support (greater than 70% ML and 0.95 PP) in our combined gene analyses (Fig 1 and S2 Fig). Ophioceraceae has a close relationship with Ceratosphaeriaceae and Pseudohalonectriaceae. However, the phylogenetic relationships of these three families are not well resolved and pending further clarification.

Phylogenetic analyses of the LSU-RPB1-ITS sequence matrix revealed that the investigated specimen (KUN-HKAS 107677) and its pure culture (KUMCC 20–0213) are grouped together and form an independent lineage basal to *Ophioceras* in Ophioceraceae with high statistical support (100% ML, 1.00 PP; Fig 1). Considering the phylogenetic results and morphology, we propose a novel species, *Ophioceras sichuanense*, occurring on submerged bamboo in Sichuan Province, China.

*Ceratosphaeria castillensis* (SMH 1865) formed a robust clade with *C. rhizomorpha* (GKM 1262) (100% ML, 1.00 PP; Fig 1) within *Ophioceras* (84% ML, 0.90 PP; Fig 1). *Ceratosphaeria castillensis* (SMH 1865) and *C. rhizomorpha* (GKM 1262) clustered with *Ophioceras aquaticus* (IFRDCC 3091, MFLUCC 16–0906), *O. dolichostomum* (CMURp50), *O. hongkongense* (HKUCC 3624), *O. submersum* (MFLUCC 18–0211) and *O. commune* (KUN-HKAS 92569, KUN-HKAS 92587, KUN-HKAS 92590, KUN-HKAS 92640) in our all analyses (Fig 1, S1 and S2 Figs) and separated distantly from taxa in Magnaporthaceae. Thus, *Ceratosphaerella* is treated as a synonym of *Ophioceras*, the prior introduced genus, in Ophioceraceae.

## Taxonomy

**Ophioceraceae** Klaubauf, Lebrun & Crous, Studies in mycology 79: 103 [1].

Type genus: *Ophioceras* Sacc.

Notes–To date, Ophioceraceae includes a single genus, *Ophioceras*. In the present phylogenetic study, Ophioceraceae formed a stable clade within Magnaporthales and distinguished from other families of Magnaporthales.

**Ophioceras** Sacc., Sylloge Fungorum 2: 358 (1883), emend. H.B. Jiang, Phookamsak & K.D. Hyde.

Facesoffungi number: FoF01255.

Synonym: *Ceratosphaerella* Huhndorf, Greif, Mugambi & A.N. Mill., Mycologia 100(6): 941 [8].

Type species: *Ophioceras dolichostomum* (Berk. & M.A. Curtis) Sacc.

*Saprobic* on bamboo, palm, bark or wood, and other herbaceous plants from aquatic or terrestrial environments. **Sexual morph**: *Ascomata* black, perithecial, immersed to superficial, scattered or gregarious, globose to subglobose, or ampulliform, glabrous with ostiolar necks, somewhat forming uni- to multi-loculate pseudostromata. *Pseudostromata if present*: locules immersed in pseudostroma, dark brown to black, subglobose to ampulliform, or irregular in shape, with a long, cylindrical, black, brittle, curved or straight, periphysate neck. *Peridium* composed of several layers, of pseudoparenchymatous cells, arranged in *textura angularis*, inner layers composed of hyaline, elongate cells, with compressed, dark brown to black cells towards the outer layers. *Paraphyses* filiform, hyaline, unbranched, septate, broad at the base, tapering at the tip. *Asci* unitunicate, 8-spored, subcylindrical to acerose or clavate, pedicellate or sessile, with a refractive J-, apical ring. *Ascospores* filiform or narrow fusiform, with rounded ends, slightly curved or sigmoidal, hyaline, pale brown or olivaceous, aseptate or septate, with or without guttulate, lacking a sheath. **Asexual morph**: Hyphomycetous,

Didymobotryum-like. *Colony on substrates* dark brown to black, rhizomorphic-like threads, radiated from central of clustered ascomata on patched subiculum. *Synnemata* formed on rhizomorphic strands, dichotomously branched hyphae, straight or flexuous, branched, lighter brown head, black in mass, with conidiophores at the apical region. *Conidiophores* elongate, septate, with dark brown bands at the septa, verrucose. *Conidiogenous cells* pale brown, cylindrical, tretic, integrated, terminal, verrucose. *Conidia* pale brown, with darker brown at the septa, ellipsoid to cylindrical, 1–3 transverse septa, verrucose [8].

Notes–*Ophioceras* occurs on a wide range of hosts mainly distributed in America, Asia, Africa and Oceania [2,5,8,9,13–17,39–53]. To date, only *O. bambusae* and *O. guttulatum* have been reported from bamboo [51,53,54]. Eleven *Ophioceras* species have been reported from freshwater [2,10], of which nine species were found in China [10,17,41,53]. In this study, *O. sichuanense* is introduced as the second species occurring on submerged decaying branches of bamboo in China.

The genus *Ophioceras* is emended herein to accommodate the genus *Ceratosphaerella* that clustered with other *Ophioceras* species in Ophioceraceae. *Ceratosphaerella* is morphologically different from *Ophioceras* in having clavate asci and hyaline to pale brown, narrow fusiform ascospores, whereas *Ophioceras* has subcylindrical to acerose asci and hyaline to olivaceous, filiform ascospores [1,8]. However, *Ophioceras* resembles *Ceratosphaerella* in the ascomatal morphology and is also supported by phylogenetic analyses. Through ML and BI phylogenetic analyses based on a concatenated LSU-RPB1-ITS sequence matrix (Fig 1), *C. castillensis* (SMH 1865), which was previously treated in Magnaporthaceae, is phylogenetically closely related to *C. rhizomorpha* and clustered within *Ophioceras* in Ophioceraceae. Therefore, we treat *Ceratosphaerella* as a synonym of *Ophioceras* instead of a genus in Ophioceraceae. Using a morpho-phylogenetic approach, *Ophioceras castillensis* comb. nov. and *O. rhizomorpha* comb. nov. are hereby introduced.

**Ophioceras sichuanense** H.B. Jiang, Phookamsak & K.D. Hyde, *sp. nov*. Fig 2.

[urn:lsid:indexfungorum.org:names:557956].

Facesoffungi number: FoF09404.

Etymology–The specific epithet "*sichuanense*" refers to Sichuan Province, P.R. China, where the species was collected.

Holotype–KUN-HKAS 107677.

*Saprobic* on decaying branches of bamboo submerged in freshwater. **Sexual morph**: *Pseudostromata* 300–750 μm diam., 230–350 μm high (excluding necks), black, scattered, solitary, semi-immersed to superficial, 1–5-loculate, glabrous, ostiolate, papillate, carbonaceous. *Locules* 150–300 μm diam., 150–250 μm high (excluding necks), immersed within pseudostroma, clustered, subglobose to ampulliform, blackened, with a long black, periphysate neck, up to 1 cm. *Peridium* 20–35 μm wide, thick-walled, composed of several layers, of flattened to broad, pseudoparenchymatous cells, arranged in *textura angularis* to *textura prismatica*, inner layers composed of hyaline cells, outer layers composed of dark brown to black pseudoparenchymatous cells. *Paraphyses* 4–8 μm wide, filiform, hyaline, indistinct septate, unbranched, slightly rough with small guttules, broad at the base, tapering toward the tip. *Asci* 90–115 × 5–6.5 μm ($\bar{x}$ = 103 × 5.8 μm, n = 20), 8-spored, unitunicate, cylindrical, sessile to subsessile, with short broad bulb-like at the base, apically rounded with a J-, apical ring. *Ascospores* 80–90 × 1–1.5 μm ($\bar{x}$ = 85 × 1.3 μm, n = 15), overlapping, or in parallel, hyaline, filiform, slightly curved to sigmoidal, thin-walled, aseptate, smooth-walled, multi-guttulate. **Asexual morph**: not observed.

Culture characteristics: Ascospores germinated on PDA within 24 hours at room temperature under normal condition. Mycelium superficial to immersed in agar medium, branched,

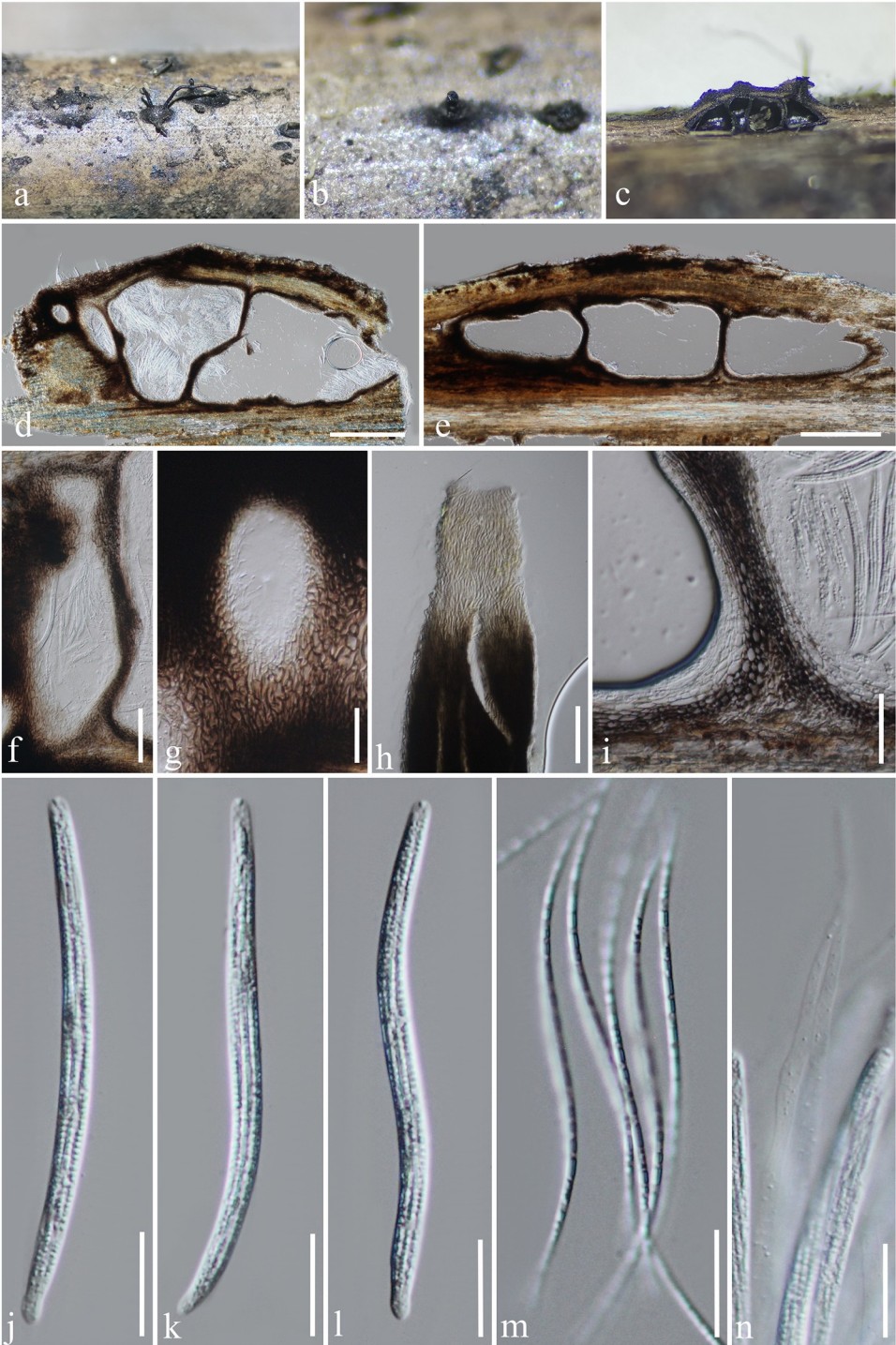

**Fig 2. *Ophioceras sichuanense* (KUN-HKAS 107677, holotype).** a, b Pseudostromata with long ostiolar necks on host. c–e Section through pseudostromata. f Ascoma. g Ostiole. h Apical part of neck. i Peridium. j–l Asci. m Ascospores. n Paraphyses. Scale bars: d, e = 200 μm, f–h = 50 μm, i = 30 μm, j–n = 20 μm.

septate, smooth hyphae. Colonies growing slowly on PDA, reaching 20 mm in two weeks, cottony, circular, with entire edge, raised, white from above and below.

Material examined: P.R. China, Sichuan Province, Yibin City, Shunan Artificial Bamboo Forest, on submerged decaying branches of bamboo, 25 July 2019, H.B. Jiang & R. Phookamsak, SC011 (KUN-HKAS 107677, holotype), ex-type living culture, KUMCC 20–0213.

GenBank accession numbers: ITS = MT995045, LSU = MT995046, SSU = MT995047 (KUMCC 20–0213; from pure culture); ITS = MW057779, LSU = MW057782, SSU = MW057847, TEF1-α = MW082017 (KUN-HKAS 107677; from fruiting bodies).

Notes–*Ophioceras sichuanense* can be distinguished from other *Ophioceras* species because it forms multi-loculate pseudostromata. *Ophioceras sichuanense* resembles *O. guttulatum*, *O. leptosporum* and *O. tenuisporum*. However, *O. sichuanense* differs from *O. guttulatum* in having smaller asci and ascospores (asci: 90–115 × 5–6.5 μm vs. 130–160 × 14–17 μm; ascospores: 80–90 × 1–1.5 μm vs. 100–128 × 4–5 μm) [17]. *Ophioceras sichuanense* resembles *O. leptosporum* and *O. tenuisporum* due to the size ranges of asci and ascospores and extremely long ostiolar necks. However, the species differs from the latter two species in their ascospore septation (Table 2). Although *O. bambusae* and *O. sichuanense* were collected from bamboos, *O. sichuanense* has longer asci and ascospores (asci: 90–115 × 5–6.5 μm vs. 90–95 × 5.5–6.5 μm; ascospores: 80–90 × 1–1.5 μm vs. 75–80 × 1.5 μm) [54]. Present phylogenetic analyses show *O. sichuanense* formed a clade basal to *Ophioceras* and close to *O. chiangdaoense* and *O. leptosporum*. *Ophioceras sichuanense* and *O. chiangdaoense* are different in the dimensions of the ostiolar neck, asci and ascospores (Table 2).

**Ophioceras castillensis** (C.L. Sm.) H.B. Jiang, Phookamsak & K.D. Hyde, *comb. nov*. Fig 3.

[urn:lsid:indexfungorum.org:names:557957].

Facesoffungi number: FoF09405.

Basionym: *Ceratosphaeria castillensis* C.L. Sm., Bull. Lab. Nat. Hist. Iowa State Univ. 2: 403 (1893).

Synonym: *Ceratosphaerella castillensis* (C.L. Sm.) Huhndorf, Greif, Mugambi & A.N. Mill., Mycologia 100(6): 944 [8].

Type information: Nicaragua, Castillo Viejo, on bark, Feb–Mar 1893, C.L. Smith, Central American Fungi 13, (isotype, NY).

Detailed description and illustration: see Huhndorf et al. [8].

Known hosts/ habitat and distribution: Saprobic on bark or wood in terrestrial habitat. To date, *Ophioceras castillensis* is only reported from Costa Rica, Nicaragua, and Puerto Rico [8].

Notes–In this study, *Ceratosphaerella castillensis* is transferred to *Ophioceras* as *O. castillensis* based on a concatenated LSU-RPB1-ITS analyses coupled with morphological similarity of the ascomata and ascomatal wall which is typical *Ophioceras*. *Ophioceras castillensis* can be separated from other *Ophioceras* species in forming ascomata on large clusters, superficial on sparse, subicular hyphae and having clavate asci and hyaline to pale brown, fusiform, 3-septate ascospores [8]. Detailed morphological comparison and taxa habitats in *Ophioceras* are described in Table 2.

**Ophioceras rhizomorpha** (Huhndorf & Mugambi) H.B. Jiang, Phookamsak & K.D. Hyde, *comb. nov*. Fig 4.

[urn:lsid:indexfungorum.org:names:557958].

Facesoffungi number: FoF09406.

Basionym: *Ceratosphaerella rhizomorpha* Huhndorf & Mugambi, Mycologia 100(6): 944 [8].

Type information: Kenya, Kagamega National Park, on decaying wood on the ground, 17 January 2007, G.K. Mugambi, GKM1262 (holotype EA, isotype F).

Detailed description and illustration: see Huhndorf et al. [8].

**Table 2. Synopsis of *Ophioceras* species.**

| Species name | Ascomata | Asci | Ascospores | Habitat | Origin | Host/substrate | References |
|---|---|---|---|---|---|---|---|
| *Ophioceras aquaticus* | 310–620 µm diam., 1-loculate, superficial to submerged, with a 500–800 µm long neck | 85–100 × 9–10 µm, cylindrical | 42–68 × 3–4 µm, filiform, slightly acute at each end, falcate, sigmoid, hyaline, 3–5-septate | Submerged | China: Yunnan | Wood | [2, 10] |
| *O. arcuatisporum* | 313–324 × 252–340 µm, 1-loculate, superficial to immersed, with a long neck, up to 800 µm | 276–307 × 15–20 µm, fusoid to narrowly cylindrical | 170–239 × 4–7 µm, narrowly fusoid to cylindrical, falcate, hyaline to pale orange, 5–12-septate | Submerged | USA: Minnesota | *Typha* sp., herbaceous debris, grass | [49] |
| *O. bambusae* | 1 mm long, 2/3 mm diam., 1-loculate, immersed, with a 2–2.5 mm long neck | 90–95 × 5.5–6.5 µm, cylindrical | 75–80 × 1.5 µm, filiform, with both blunt ends, curved, hyaline, indistinctly septate | Terrestrial | Indonesia: Java | Bamboo | [54] |
| *O. castillensis* | 525–650 µm diam., 1-loculate, superficial, with a 250–400 µm long neck | 70–90 × 10–14 µm, clavate | 29–40 × 4–5.5 µm, narrowly fusiform, slightly curved, hyaline to pale brown, 3-septate | Terrestrial | Nicaragua | Bark and wood | [8] |
| *O. cecropiae* | 200–250 µm diam., 1-loculate, immersed, with a long neck, up to 2 mm | 75–90 × 6.5–7.5 µm, cylindrical to subfusoid | 60–70 × 2 µm, filiform, straight to slightly curved, hyaline, septate | Terrestrial | Venezuela | *Cecropia* sp. | [55] |
| *O. chiangdaoense* | 200–310 × 170–310 µm, 1-loculate, immersed, with a 93–273 µm long neck | 85–125 × 11–17 µm, cylindrical | 54–75.5 × 4–5.5 µm, filiform, falcate, fusoid at both ends, hyaline, 3-septate | Terrestrial | Thailand: Chiang Mai | Decaying leaves of *Dracaena loureiroi* | [9] |
| *O. commune* | 150–350 × 260–400 µm, 1-loculate, immersed to superficial, with a 375–1660 µm long neck | 64–118 × 4–12 µm, cylindrical | 50–110 × 2 µm, filiform, arcuate or sigmoidal, hyaline, 3–7-septate | Submerged | Panama: Barro Colorado Island | Wood, herbaceous debris | [2, 49] |
| *O. dolichostomum* | 500 µm diam., 1-loculate, immersed, with a 1–5 mm long neck | 100–130 × 8–12 µm, cylindrical | 94–110 × 2–3 µm, filiform, falcate, sigmoid, hyaline, 3–7-septate | Submerged | USA: Florida | Dead wood | [2, 46] |
| *O. filiforme* | 3–5 mm long, 150–180 µm diam., 1-loculate, immersed, erumpent to superficial, with a long neck | 100–120 × 10–13 µm, clavate or fusoid | 80–100 × 3–3.5 µm, filiform, hyaline to yellowish, multi-septate | Terrestrial | Indonesia: Java | Rotten leaf sheaths of *Amomum* sp. | [56] |
| *O. fusiforme* | 360–500 × 330–450 µm, 1-loculate, immersed to erumpent, with a 250–800 µm long neck | 70–112 × 6–12 µm, cylindrical | 64–104 × 1.5–3 µm, filiform, fusoid, tapering at both ends, straight to falcate, 3–5-septate | Submerged | USA: Indiana | Decorticated woody debris | [49] |
| *O. guttulatum* | 400–600 × 1200–1800 µm, 1-loculate, superficial to immersed, with a 500–1500 µm long neck | 130–160 × 14–17 µm, broadly cylindrical | 100–128 × 4–5 µm, cylindrical, falcate, pale yellow to hyaline, 3–5-septate | Submerged | China: Hong Kong | Wood | [17] |
| *O. hongkongense* | 500–640 × 700–800 µm, 1-loculate, superficial to immersed, with a more than 600 µm long neck | 100–125 × 12–14 µm, elongated fusoid to broadly cylindrical | 72–101 × 3.5–4.5 µm, cylindrical, falcate, tapered at both ends, hyaline, 3–5-septate | Submerged | China: Hong Kong | Wood | [17] |
| *O. indicus* | 400–650 µm diam., 1-loculate, immersed, with a long neck, up to 1.5 mm | 65–90 × 8.5–11.5 µm, cylindrical to subfusoid | 60–85 × 2.5–3.5 µm, filiform, tapering towards base, slightly curved, hyaline to subhyaline, 7–10-septate | Terrestrial | India: New Delhi | Dried twigs of *Ficus infectoria* | [47] |
| *O. leptosporum* | 250–300 µm diam., 1-loculate, immersed or superficial, with a 1–2 mm long neck | 70–95 × 5–6 µm, cylindrical | 70–80 × 1–1.5 µm, filiform, apex rounded, base acute, hyaline to faintly tinted, straight to slightly curved or sigmoid, 3–7-septate | Submerged | UK: Exeter | Rotten stems of *Umbelliferae* sp. | [4] |
| *O. miyazakiense* | Data unavailable | Data unavailable | Data unavailable | Terrestrial | Japan: Kyushu | Decaying litter | [57] |

(*Continued*)

**Table 2.** (Continued)

| Species name | Ascomata | Asci | Ascospores | Habitat | Origin | Host/substrate | References |
|---|---|---|---|---|---|---|---|
| *O. palmae* | 164–320 × 244–288 μm, 1-loculate, partly immersed, with a 180–376 μm long neck | 76–96 × 10–14 μm, broadly cylindrical | 79–90 × 1.2–2 μm, filiform, tapering at both ends, sigmoid, hyaline, 5-septate | Terrestrial | Philippines: Mt. Makiling | *Calamus ornatus* | [17] |
| *O. parasiticum* | 600–800 μm diam. | 100–140 × 9–11 μm | 48–70 × 2.5–3.3 μm, 3–9-septate | Terrestrial (parasite) | China | Data unavailable | [9] |
| *O. petrakii* | 600–750 × 555–675 μm, 1-loculate, immersed, with a neck | 171–182 × 12–15 μm, cylindrical | 152–171 × 3–4 μm, filiform, apex rounded, base acute, slightly curved or sigmoid, hyaline, multi-septate | Terrestrial | India: Karnataka | Dead stems of *Vitex negundo* | [45] |
| *O. rhizomorpha* | 500–900 × 500–750 μm, 1-loculate, superficial, with a 300–600 μm long neck | 115–145 × 13–16 μm, clavate | 39–49 × 3.5–4.5 μm, narrowly fusiform, slightly curved, hyaline to pale brown, 3-septate | Terrestrial | Kenya | Bark or wood | [8] |
| *O. sichuanense* | 230–350 × 300–750 μm, pseudostromatic, 1–5-loculate, semi-immersed to superficial, with long necks, up to 1 cm | 90–115 × 5–6.5 μm, cylindrical | 80–90 × 1–1.5 μm, filiform, slightly curved to sigmoidal, hyaline, aseptate | Submerged | China: Sichuan | Decaying branches of bamboo | This study |
| *O. sorghi* | 300–400 μm diam., 1-loculate, immersed, with a 350–700 μm long neck | 85–110 × 12–14 μm, cylindrical to clavate | 75–95 × 3–4 μm, filiform, cylindrical, with rounded apex and slightly thinner rounded base, slightly curved, hyaline, 3–12-septate | Terrestrial | Central African Republic: M'Baiki | *Sorghum vulgare* | [4, 43] |
| *O. submersum* | 300–400 × 500–600 μm, 1-loculate, immersed, with a long neck | 115–137 × 10–11 μm, cylindrical | 87–109 × 3–4 μm, filiform, rounded at both ends, slightly curved or sigmoid, hyaline, multi-septate | Submerged | Thailand | Wood | [2] |
| *O. tambopataense* | Data unavailable | Data unavailable | Data unavailable | Terrestrial | Peru | Decaying leaf of palm | [57] |
| *O. tenuisporum* | 240–625 × 260–775 μm, 1-loculate, superficial to partially immersed, with a long neck, up to 20 mm | 82–114 × 4–6 μm, cylindrical to narrowly fusoid | 66–94 × 1–1.5 μm, filiform, more broadly rounded at one end than the other, curved to sigmoid, hyaline, 3-septate | Submerged | Panama: Barro Colorado Island | Twig | [49] |
| *O. venezuelense* | 730–890 × 745–868 μm, 1-loculate, partially immersed to superficial, with a 250–800 μm long neck | 148–180 × 11–18 μm, cylindrical to narrowly fusoid | 130–158 × 2–4 μm, filiform, falcate, more broadly rounded at one end than the other, straight to slightly curved, hyaline, (4)–5–(6)-septate | Submerged | Venezuela: Portuguesa | Wood, herbaceous debris | [49] |
| *O. zeae* | 450–650 μm diam., 1-loculate, superficial or partially immersed, with a 350–900 μm long neck | 55–65 × 7–8 μm, cylindrical to narrowly fusoid | 39–50 × 2.3–2.5 μm, cylindrical to fusoid, with rounded ends, straight or slightly curved, hyaline, 3-septate | Terrestrial | Central African Republic: Boukoko | Dead *Zea mays* | [4, 42] |

Known hosts/ habitat and distribution: Saprobic on decaying wood of terrestrial habitat. To date, *Ophioceras rhizomorpha* is only reported from Kenya.

Notes–*Ophioceras rhizomorpha* was reported with a Didymobotryum-like asexual morph on a natural substrate [8]. The species resembles *O. castillensis* in forming ascomata on the large subicular, and having clavate asci and hyaline to pale brown, fusiform, 3-septate ascospores [8]. *Ophiocera rhizomorpha*, however, differs from *O. castillensis* in having larger ascomata, necks, peridia, paraphyses, asci and ascospores [8] (see Table 2).

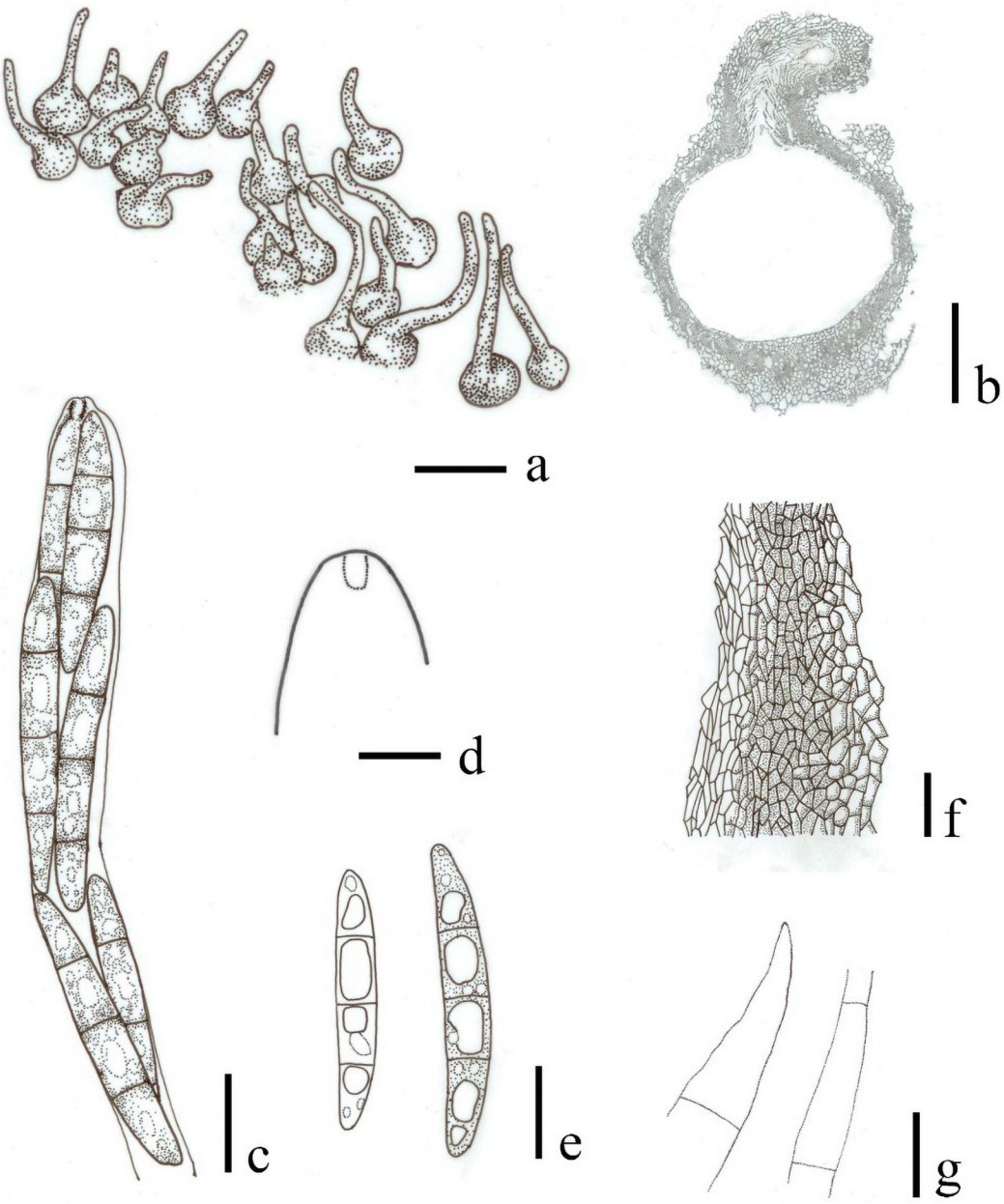

**Fig 3. *Ophioceras castillensis* (redrawn from Huhndorf et al. [8], NY isotype).** a, b Ascomata. c Ascus. d Apical ring. e Ascospores. f Peridium. g Paraphyses. Scale bars: a = 1 mm, b = 200 μm, f = 20 μm, c, e, g = 10 μm, d = 5 μm.

## Discussion

Ophioceraceae currently accommodates only *Ophioceras*. Maharachchikumbura et al. [58] accommodated the family in Magnaporthales (Diaporthomycetidae, Sordariomycetes) based on literature review and phylogenetic analysis. Ophioceraceae has limited taxon sampling, and most taxa in this family lack reliable protein coding genes to clarify phylogenetic affinities. For example, *Ophioceras arcuatisporum* (strains A9-1, A167-1B) has only SSU sequence data

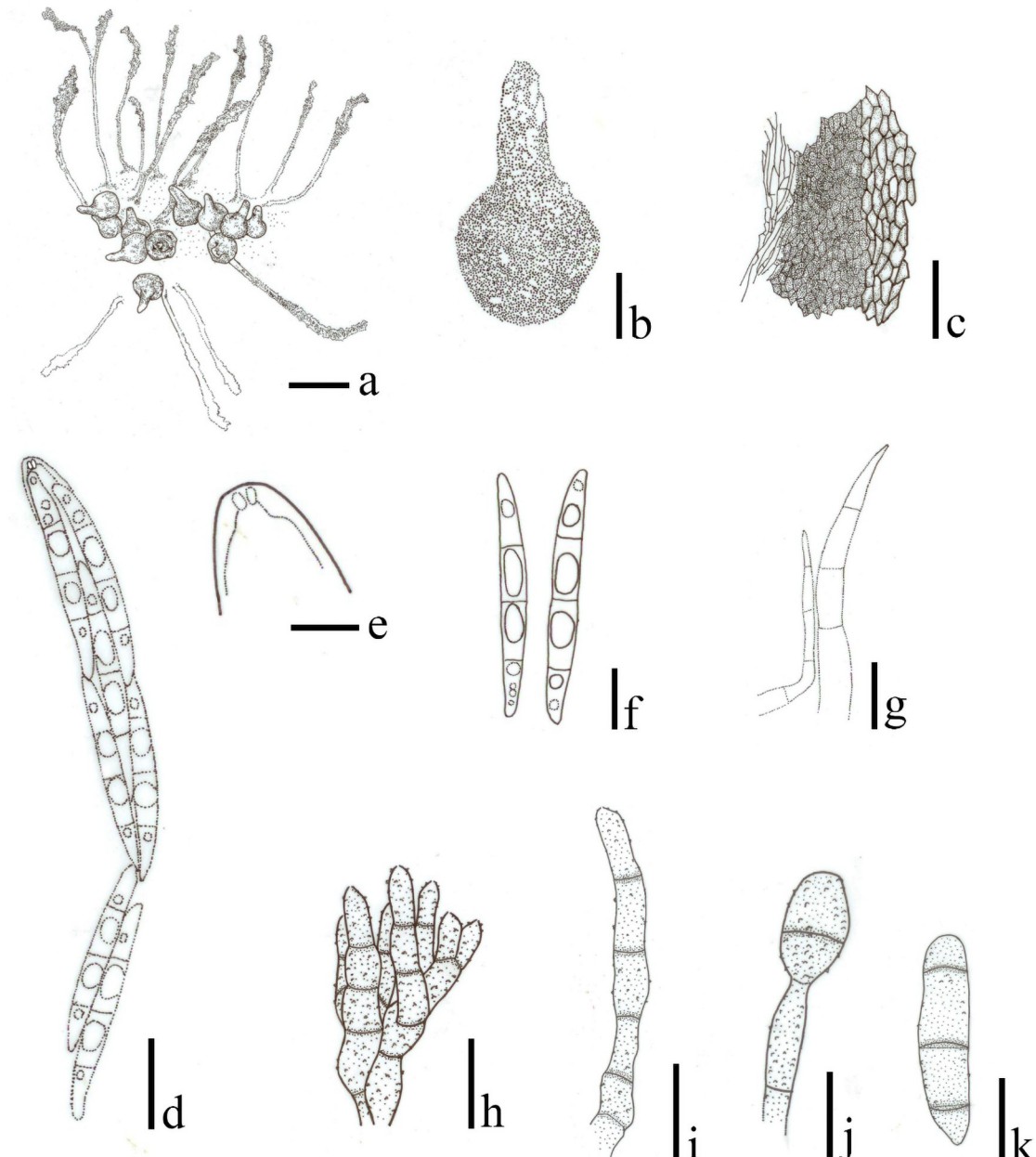

**Fig 4. *Ophioceras rhizomorpha* (redrawn from Huhndorf et al. [8], all from Mugambi 1262).** a Ascomata and synnemata on the substrate. b Ascoma. c Peridium. d Ascus. e Apical ring. f Hyaline to pale brown ascospores. g Paraphyses. h, i Conidiophores. j Conidiogenous cell bearing conidium. k Conidium. Scale bars: a = 1 mm, b = 300 μm, c = 50 μm, g = 30 μm, d = 20 μm, i, k = 15 μm, f, h, j = 10 μm, e = 5 μm.

available in GenBank. In preliminary phylogenetic analysis of SSU sequence matrix, the species formed a stable clade within Magnaporthaceae rather than Ophioceraceae. *Ophioceras arcuatisporum* needs to be re-visited and re-illustrated, incorporating details from molecular data.

Luo et al. [2] performed combined LSU and TEF1-α phylogenetic analyses to investigate the relationships of taxa in Magnaporthales. In their phylogeny, *Ceratosphaeria* grouped with

*Pseudohalonectria* (Pseudohalonectriaceae) and separated from *Ophioceras* (Ophioceraceae). Based on the fact that *Ceratosphaeria* differs from *Pseudohalonectria* in having narrow cylindric-fusiform to filiform and longer ascospores, Ceratosphaeriaceae was thus introduced as a new family within Magnaporthales to accommodate *Ceratosphaeria* [2]. In the present study, we performed an updated phylogenetic tree based on a concatenated LSU-RPB1-ITS sequences and showed that *Ceratosphaeria* (Ceratosphaeriaceae) clustered with *Ophioceras* (Ophioceraceae) with low statistical support, suggesting that gene selection in the data matrix affects the tree topology at the familial levels in Magnaporthaceae. *Ceratosphaeria* is morphologically similar to *Ceratosphaerella* [8]. Although *Ceratosphaeria* clustered with *Ophioceras* with low statistical support, *Ceratosphaeria* possibly belongs to *Ophioceraceae*. However, the phylogenetic status of *Ceratosphaeria* needs to be clarified with more evidence in the future studies.

In the present study, we synonymize *Ceratosphaerella* under *Ophioceras* based on molecular phylogeny coupled with similar ascomatal morphology. Phylogenetic analyses based on the LSU sequence dataset (S1 Fig) and the concatenated LSU-ITS (S2 Fig) and the LSU-RPB1-ITS (Fig 1) sequence datasets have always shown that *O. castillensis* (≡ *C. castillensis*) and *O. rhizomorpha* (≡ *C. rhizomorpha*) formed a stable clade within *Ophioceras*. *Ophioceras castillensis*, *O. rhizomorpha* and most *Ophioceras* species lack protein coding genes and other reliable genes to clarify phylogenetic placement as well as limited taxon sampling. The ex-type strain of *O. rhizomorpha* and the reference strain of *O. castillensis* were sequenced only for LSU locus. Hence, more reliable gene loci (e.g., SSU, ITS, RPB1 and TEF1-α) from the ex-type strain of *O. rhizomorpha* should be obtained and the epitype of *O. castillensis* should be designated and incorporated with morpho-molecular based taxonomic treatment. Furthermore, the new collections and sequence data of taxa in *Ophioceras* are required to provide a better taxonomic resolution for robust species delineations in this genus.

Many genera in Magnaporthales have similar morphological characteristics with *Ophioceras* (Table 3). However, these genera are considered distinct genera based on phylogenetic investigations [1,2,8,59–61]. *Pseudohalonectria* (Pseudohalonectriaceae) is also similar to

**Table 3. Morphological comparisons of similar genera to *Ophioceras*.**

| Generic name | Ascomata | Asci | Ascospores | References |
|---|---|---|---|---|
| *Aquafiliformis* (Magnaporthaceae) | Globose to subglobose, with long beak | Cylindrical to clavate, with an inconspicuous apical ring | Aseptate, filiform, hyaline | [2] |
| *Ceratosphaeria* (Ceratosphaeriaceae) | Globose to pyriform, with a long black or yellow neck | Cylindrical, with a conspicuous, non-amyloid, apical ring | Multi-septate, filiform, hyaline | [2] |
| *Gaeumannomycella* (Magnaporthaceae) | Subglobose to elliptical, with a lateral, central cylindrical neck | Cylindrical to elongated clavate, apical ring not observed | 0–3-septate, narrowly fusiform, hyaline | [61] |
| *Gaeumannomyces* (Magnaporthaceae) | Subglobose to elliptical, with a cylindrical neck | Cylindrical to elongated clavate, with an apical refringent ring | Indistinctly septate, filiform, hyaline to pale brown | [60] |
| *Kohlmeyeriopsis* (Magnaporthaceae) | Ellipsoid, with a long cylindrical periphysate neck | Fusoid to cylindrical, with a large apical ring | Indistinctly septate, filamentous, tapering towards the base, hyaline | [1, 63] |
| *Muraeriata* (Magnaporthaceae) | Lageniform to globose, with long beak | Cylindrical to ventricose, with a tall, narrow, apical ring | 3-septate, narrowly fusiform, ends slightly curved, hyaline | [8] |
| *Neogaeumannomyces* (Magnaporthaceae) | Globose to subglobose, with a long, periphysate neck | Cylindrical, with an apical ring | 2–3-septate, filiform to long fusiform, hyaline | [59] |
| *Ophioceras* (Ophioceraceae) | Subglobose to ampulliform, with a long cylindrical periphysate neck | Subcylindrical to narrowly fusoid or clavate asci with a small, refractive, non-amyloid apical ring | Aseptate to multi-septate, filiform, narrowly fusiform, hyaline pale brown or olivaceous | [2, 8, this study] |
| *Slopeiomyces* (Magnaporthaceae) | Globose, with periphysate neck bearing hyphae | Clavate, with a non-amyloid apical ring | 3–4-septate cylindrical to fusoid, tapering somewhat to base, hyaline | [1] |

*Ophioceras* in its ascomata and asci. However, *Pseudohalonectria* varied in shape of ascospores, such as ellipsoidal, fusiform or filiform [62]. In the present study, *Pseudohalonectria* (Pseudohalonectriaceae) formed an independent lineage separate from other families in Magnaporthales; however, *Pseudohalonectria* could not be resolved at the species level such as in Perera et al. [62]. It may be because molecular data of most taxa in this genus are unavailable in GenBank database. Moreover, some sequences of *P. lignicola* deposited in GenBank are likely to be misidentified [62]. Therefore, sequences of *Pseudohalonectria* species used for phylogenetic analyses are limited.

## Supporting information

**S1 Fig. RAxML tree generated from an analysis of the LSU matrix of taxa in Magnaporthales.**
(TIF)

**S2 Fig. RAxML tree generated from an analysis of the LSU-ITS matrix of taxa in Magnaporthales.**
(TIF)

## Acknowledgments

The authors gracefully thank the Biology Experimental Center, Germplasm Bank of Wild Species, Kunming Institute of Botany, Chinese Academy of Sciences for providing the facilities of molecular laboratory. We also thank Shaun Pennycook from Manaaki Whenua—Landcare Research for his assistance in naming the new species in the genus *Ophioceras*. Austin G. Smith is thanked for the English proofreading of this manuscript. Hong-Bo Jiang would like to thank Mae Fah Luang University for Ph.D scholarship.

## Author Contributions

**Conceptualization:** Hong-Bo Jiang, Kevin D. Hyde, Rungtiwa Phookamsak, Saisamorn Lumyong.

**Data curation:** Hong-Bo Jiang.

**Formal analysis:** Hong-Bo Jiang, Rungtiwa Phookamsak.

**Funding acquisition:** Ali H. Bahkali, Abdallah M. Elgorban, Samantha C. Karunarathna, Rungtiwa Phookamsak, Saisamorn Lumyong.

**Investigation:** Hong-Bo Jiang, Kevin D. Hyde, Er-Fu Yang, Rungtiwa Phookamsak.

**Methodology:** Hong-Bo Jiang, Rungtiwa Phookamsak.

**Project administration:** Rungtiwa Phookamsak.

**Supervision:** Kevin D. Hyde, Pattana Kakumyan, Rungtiwa Phookamsak, Saisamorn Lumyong.

**Writing – original draft:** Hong-Bo Jiang, Er-Fu Yang, Rungtiwa Phookamsak.

**Writing – review & editing:** Kevin D. Hyde, Pattana Kakumyan, Samantha C. Karunarathna, Saisamorn Lumyong.

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
