## [Decision Letter · Decision Letter 0]

19 Mar 2021

PONE-D-20-37408

Morphological and phylogenetic appraisal of Ophioceras (Ophioceraceae, Magnaporthales)

PLOS ONE

Dear Dr. Phookamsak,

Thank you for submitting your manuscript to PLOS ONE. After careful consideration, we feel that it has merit but does not fully meet PLOS ONE’s publication criteria as it currently stands. Therefore, we invite you to submit a revised version of the manuscript that addresses the points raised during the review process.

I would like to sincerely apologise for the delay you have incurred with your submission. It has been exceptionally difficult to secure reviewers to evaluate your study. We have now received two completed reviews; their comments are available below.

Both reviewers have raised some concerns about the study that need to be addressed in a revision. In particular, please consider Reviewer#2 comments and ensure that all the data deposited on treebase is fully available.

Please revise the manuscript to address all the reviewer's comments in a point-by-point response in order to ensure it is meeting the journal's publication criteria. Please note that the revised manuscript will need to undergo further review, we thus cannot at this point anticipate the outcome of the evaluation process.

We look forward to receiving your revised manuscript.

Kind regards,

Miquel Vall-llosera Camps

Senior Editor

PLOS ONE

Journal Requirements:

Reviewers' comments:

Reviewer's Responses to Questions

**Comments to the Author**

1. Is the manuscript technically sound, and do the data support the conclusions?

Reviewer #1: Yes

Reviewer #2: Yes

2. Has the statistical analysis been performed appropriately and rigorously? 

Reviewer #1: Yes

Reviewer #2: N/A

3. Have the authors made all data underlying the findings in their manuscript fully available?

Reviewer #1: Yes

Reviewer #2: No

4. Is the manuscript presented in an intelligible fashion and written in standard English?

Reviewer #1: Yes

Reviewer #2: Yes

5. Review Comments to the Author

Reviewer #1: Remove description from the abstract. The intro already starts with it.

166-168 Remove The from "The USA". I suggest double checking the use of "the" across the text. There are some sentences that sound a bit odd, but nothing particularly problematic.

Fig1. Try to convert to png. Jpg files lose resolution. Blue characters with green background might be difficult to see. I suggest inverting the colors (yellow-green). Tree is missing support values for some nodes, specially the node that separates Pseudohalonectriaceae and Ceratosphaeraceae+Ophioceraceae.

249 Didymobotryum should be capitalized, as it refers to a genus. Also, I think it would be adequate to describe what Didymobotryum-like means.

Overall great job. Other than these minor complaints I see no reason to publish this work.

Reviewer #2: The manuscript presents new information for the genus Ophiocera based on phylogenetic and morphological data, including a new species and two new combinations. The data presented is relevant and well presented, but some points must be clarified. Also, I could not access the treebase archives and thus, can not ensure that the data is available. I'm attaching a file with my comments and questions in the text.

6. PLOS authors have the option to publish the peer review history of their article (what does this mean?). If published, this will include your full peer review and any attached files.

Reviewer #1: No

Reviewer #2: No

---

## [Author Response · Author response to Decision Letter 0]

10 Apr 2021

We have revised our manuscript based on the reviewer's comments and hope that the updated version is satisfied to published in Plos One.

---

## [Decision Letter · Decision Letter 1]

11 May 2021

PONE-D-20-37408R1

Morphological and phylogenetic appraisal of Ophioceras (Ophioceraceae, Magnaporthales)

PLOS ONE

Dear Dr. Phookamsak,

Thank you for submitting your manuscript to PLOS ONE. After careful consideration, we feel that it has merit but does not fully meet PLOS ONE’s publication criteria as it currently stands. Therefore, we invite you to submit a revised version of the manuscript that addresses the points raised during the review process.

We look forward to receiving your revised manuscript.

Kind regards,

Tamás Papp, PhD

Academic Editor

PLOS ONE

Journal Requirements:

Reviewers' comments:

Reviewer's Responses to Questions

**Comments to the Author**

1. If the authors have adequately addressed your comments raised in a previous round of review and you feel that this manuscript is now acceptable for publication, you may indicate that here to bypass the “Comments to the Author” section, enter your conflict of interest statement in the “Confidential to Editor” section, and submit your "Accept" recommendation.

Reviewer #1: All comments have been addressed

Reviewer #2: (No Response)

2. Is the manuscript technically sound, and do the data support the conclusions?

Reviewer #1: Yes

Reviewer #2: Yes

3. Has the statistical analysis been performed appropriately and rigorously? 

Reviewer #1: Yes

Reviewer #2: N/A

4. Have the authors made all data underlying the findings in their manuscript fully available?

Reviewer #1: Yes

Reviewer #2: No

5. Is the manuscript presented in an intelligible fashion and written in standard English?

Reviewer #1: Yes

Reviewer #2: Yes

6. Review Comments to the Author

Reviewer #1: Please be sure that the images are not in JPEG, it's a format with very low quality and makes it difficult to zoom in.

Reviewer #2: Dear authors

please check the comments in the text.

I reinforce that you should provide a link for checking the matrices and topologies in treebase. Fully availablility of the results is a request of Plos One.

7. PLOS authors have the option to publish the peer review history of their article (what does this mean?). If published, this will include your full peer review and any attached files.

Reviewer #1: No

Reviewer #2: No

---

## [Author Response · Author response to Decision Letter 1]

25 May 2021

Dear Reviewer,

Thank you very much for your valuable comments which are highly insightful and enable us to greatly improve the quality of our manuscript. We have revised our manuscript point by point following your comments and hopefully, this version is satisfied to published. If there is more correction, please let us know. 

Yours sincerely,

Rungtiwa Phookamsak

---

## [Decision Letter · Decision Letter 2]

15 Jun 2021

Morphological and phylogenetic appraisal of Ophioceras (Ophioceraceae, Magnaporthales)

PONE-D-20-37408R2

Dear Dr. Phookamsak,

We’re pleased to inform you that your manuscript has been judged scientifically suitable for publication and will be formally accepted for publication once it meets all outstanding technical requirements.

Kind regards,

Tamás Papp, PhD

Academic Editor

PLOS ONE

Additional Editor Comments (optional):

Reviewers' comments:

Reviewer's Responses to Questions

**Comments to the Author**

1. If the authors have adequately addressed your comments raised in a previous round of review and you feel that this manuscript is now acceptable for publication, you may indicate that here to bypass the “Comments to the Author” section, enter your conflict of interest statement in the “Confidential to Editor” section, and submit your "Accept" recommendation.

Reviewer #1: All comments have been addressed

Reviewer #2: All comments have been addressed

2. Is the manuscript technically sound, and do the data support the conclusions?

Reviewer #1: Yes

Reviewer #2: Yes

3. Has the statistical analysis been performed appropriately and rigorously? 

Reviewer #1: Yes

Reviewer #2: (No Response)

4. Have the authors made all data underlying the findings in their manuscript fully available?

Reviewer #1: Yes

Reviewer #2: Yes

5. Is the manuscript presented in an intelligible fashion and written in standard English?

Reviewer #1: Yes

Reviewer #2: Yes

6. Review Comments to the Author

Reviewer #1: (No Response)

Reviewer #2: (No Response)

7. PLOS authors have the option to publish the peer review history of their article (what does this mean?). If published, this will include your full peer review and any attached files.

Reviewer #1: No

Reviewer #2: No

---

## [Editor Report · Acceptance letter]

2 Aug 2021

PONE-D-20-37408R2 

Morphological and phylogenetic appraisal of *Ophioceras* (Ophioceraceae, Magnaporthales) 

Dear Dr. Phookamsak:

I'm pleased to inform you that your manuscript has been deemed suitable for publication in PLOS ONE. Congratulations! Your manuscript is now with our production department. 

Kind regards, 

on behalf of

Dr. Tamás Papp 

Academic Editor

PLOS ONE